# Knowledge and preventive barriers towards conducting systematic review among undergraduate medical students of Arab countries: A multi country online survey

Elfatih A. Hasabo[1,2,3,4*‡], Walaa Elnaiem[1,2‡], Alaa S. Ahmed[1,2], Azza E. A. Abdalla[2,5], Khabab Abbasher Hussien Mohamed Ahmed[1,2], Ghassan Elfatih Mustafa Ahmed[1,2], Mohamed Sati Shampool Abdelgader[1,2], Alamin Alfatih[1,2], Huda A. Sherif[2,6], Omar Al Komi[2,7], Hajar Alkokhiya Aldare[2,8], Amira Yasmine Benmelouka[2,9], Afnan W. M. Jobran[2,10], Tayba A. Mugibel[2,11], Mohamad Imad Al-Kassih[2,7], Muhamad Zakaria Brimo Alsaman[2,12], Ahmed Aljabali[2,13], Mohammed Mahmmoud Fadelallah Eljack[2,14], Sudan Analytics Research Group team of collaborators¶

1 Faculty of Medicine, University of Khartoum, Khartoum, Sudan, 2 Sudan Analytics Research Group, Khartoum, Sudan, 3 Royal College of Surgeons in Ireland (RCSI) University of Medicine and Health Sciences, Dublin, Ireland, 4 Precision Cardiovascular Medicine & Innovation Institute (PCMI), Cardiovascular Research Institute Dublin (CVRI), Mater Private Network, Dublin, Ireland, 5 Department of Internal Medicine, Rochester General Hospital, New York, United States of America, 6 Faculty of Medicine, University of Menofia, Menofia, Egypt, 7 College of Medicine, Sulaiman Alrajhi University, Albukayriyah, Al-Qassim, Saudi Arabia, 8 Faculty of Medicine, Sebha University, Sabha, Libya, 9 Faculty of Medicine, University of Algiers, Algiers, Algeria, 10 Faculty of Medicine, Al Quds University, Jerusalem, Palestine, 11 College of Medicine, Hadhramout University, Hadhramout, Yemen, 12 Vascular Surgery Department, Al-Razi Hospital, Aleppo, Syria, 13 Faculty of Medicine, Jordan University of Science and Technology, Irbid, Jordan, 14 Faculty of Medicine, University of Bakht Alruda, Ed Dueim, Sudan

¶ Membership of the Sudan Analytics Research Group team of collaborators is provided in the Acknowledgments.
‡These author has an equal contribution, and both were the first author
* elfatih.ahmed.hasabo@gmail.com

## Abstract

### Background

Systematic reviews (SR) provide the highest level of evidence in research. Medical students are encouraged to learn how to conduct SR, yet barriers to engaging in these reviews need to be identified to enhance their implementation. This study aimed to assess the knowledge, practices, and perceived barriers to conducting SR among undergraduate medical students from Arab countries.

### Methods

A cross-sectional study was conducted involving undergraduate medical students from nine Arab countries enrolled in public and private medical schools. Sociodemographic information, as well as data on knowledge and barriers to conducting SR, were collected from participants through an online survey. The level of knowledge regarding SR was measured using a set of questions, with a total score of 19.

**Data availability statement:** The datasets used in this study are available from the following DOI link: https://doi.org/10.6084/m9.figshare.24481459.

**Funding:** The author(s) received no specific funding for this work.

**Competing interests:** The authors have declared that no competing interests exist.

Adjusted odds ratio (AOR) were used to find the associated factors with good knowledge of SR.

## Results

With a response rate of 89.7%, 13,060 participants were enrolled, of whom 58.9% were female and 77.0% were studying at public universities. Additionally, 49.0% were in their clinical years. Approximately 31% had heard about SR, and 3,275 participants (25.1%) had attended training on SR. Overall, only 4.3% of participants demonstrated good knowledge of SRs. Multivariate logistic regression analysis revealed that age (AOR = 1.111, 95% CI: 1.069–1.154) and participation in research-related activities (AOR = 4.501, 95% CI: 3.650–5.551) were significantly associated with good knowledge of SR. The most identified barriers to conducting SRs included a lack of knowledge about SR (47.0%) and a lack of research exposure and opportunities (28.8%). Regarding engagement in secondary research, only 1,567 participants (12.0%) had participated in a secondary research project, and of those, only 471 (30.1%) had published their work. The types of enrolled research projects included SR (62.3%), systematic reviews with meta-analysis (43.3%), and network meta-analysis (33.4%).

## Conclusion

The findings indicate a poor level of knowledge regarding SR among participants and highlight several barriers preventing undergraduate medical students from engaging in this research. There is a pressing need for further training on SR to enhance the knowledge and practice of SR among undergraduate medical students.

## Introduction

The systematic review (SR) is a qualitative synthesis of information addressing a specific research question. It is conducted following a prespecified protocol to objectively select articles for inclusion. The concept of SR was first introduced in the early 1990s as part of evidence-based medicine (EBM) to bridge the growing gap between research and practice. Since then, SRs have become one of the most popular forms of research reviews, with over 15,000 publications indexed as SR in PubMed by 2018 [1–3]. The SR may include a meta-analysis when sufficient data is available, using statistical tools to provide a quantitative synthesis of the literature. Importantly, a meta-analysis is a statistical procedure that pools the effect sizes of all included studies within an SR to produce an objective evaluation of the information based on numerical data [4]. Systematic reviews serve multiple functions, such as keeping clinicians informed about specific topics, assessing the methodological quality of existing studies, identifying knowledge gaps, and reconciling conflicting findings in the literature [5].

Applications and conducting SR, make it the gold standard in clinical research, particularly for the synthesis of clinical trials. When conducted correctly, SR can save

resources by reducing the need for redundant clinical trials [6]. However, conducting SR requires a thorough understanding of the methodology, which can be challenging. This is especially true in developing countries where research mentorship is limited [7]. The primary goal of SR is to systematically identify and synthesize all relevant studies on a specific health-related topic, making the information accessible to decision-makers and other stakeholders [4].

Systematic reviews also provide a rigorous method for evaluating the relative effectiveness of interventions and guiding clinical practice. Despite the significant value of SR, recent literature has raised concerns about biases and misleading conclusions, often stemming from a lack of standardized definitions for secondary research methodologies or overly broad generalizing the results of SR [2,8].

Several barriers hinder the uptake of SR, including financial constraints, the overwhelming volume of primary research, insufficient guidance for local adaptation of globally synthesized evidence, limited critical appraisal skills, poor awareness of evidence sources, and time limitations for researchers [9].

To date, no study has specifically assessed the knowledge, practices, and barriers to conducting SR among under-graduate medical students in Arab countries. Furthermore, scientific data on secondary research and systematic review among medical students globally are also lacking. This study aims to address this gap by examining the knowledge and preventive barriers toward conducting SR among undergraduate medical students in Arab countries, as well as the factors associated with good knowledge.

## Methods

### Study design and settings

This cross-sectional study was conducted between January and June 2022, during the COVID-19 pandemic, using an online survey. The survey aimed to measure the level of knowledge, practices, and perceived barriers to conducting SR among undergraduate medical students across public and private universities in nine Arab countries: Algeria, Egypt, Jordan, Libya, Palestine, Saudi Arabia, Sudan, Syria, and Yemen. Collaborators from the included countries distributed the online survey to students at the selected universities.

### Participants

All undergraduate medical students with access to the internet and above the age of 18 and enrolled in one of the selected universities were invited to participate. Students unwilling to participate were excluded from the study.

### Instruments used to measure the variables of interest in the study

The survey questionnaire was designed after an extensive literature review and converted into an online Google Form to collect data on the following variables:

1. **Sociodemographic information**: Including age, gender, residence, marital status, type of university, year of study, presence of relatives in the healthcare field, and prior participation in research-related activities.

2. **Knowledge about SR**: Assessed via 19 multiple-choice questions. Each correct response was scored as one point, and the maximum score was 19. A score of ≥50% indicated good knowledge, while a score of <50% indicated poor knowledge.

3. **Awareness of medical databases**: A list of medical databases was provided for the participants, and they were asked to identify the databases they knew.

4. **Skills for systematic review**: Proficiency across 11 key steps of conducting SR and MA was self-rated using a 5-point Likert scale (1 = poor, 5 = advanced).

5. **Perceived barriers**: Participants highlighted barriers to conducting SRs from a predefined list.

6. **Practice of secondary research**: Previous participation in research, types of research projects, roles, and databases or software used, as well as publication outcomes, were analyzed.

Before distributing the questionnaire to participants, it underwent face validity checks and pilot testing to ensure clarity. The internal consistency of the questionnaire was measured using Cronbach's alpha and found to be ≥ 0.80 for the knowledge and perceived barriers sections, and ≥ 0.90 for the rating skills of systematic review and practice sections.

### Survey administration and sampling

Collaborators from included countries were recruited using a Google form after posting an announcement for recruitment by Sudan Analytics Association on social media (Telegram, Facebook, LinkedIn, among other platforms). Convenient sampling was used in this study to include every participant who wanted to fill in the online survey because of the inability to conduct a random sampling due to the COVID −19 pandemic and many the included countries and universities. Collaborators distributed the online Google form among the students in their universities, and they sent regular reminders during the study period for the students to participate.

### Data management for quality assurance

To ensure the quality of the data, collaborators from different countries were included in this study. The collaborators invited students to participate in the study. Several students in each country agreed to participate. Then, the online survey was distributed to the participants via Google Forms. Data associated with this manuscript was uploaded as a supplementary file.

### Ethics approval and consent to participate

The ethical approval of this study (Approval no: 4–22, 1/2/2022) was obtained from the Health-Sector Ethical Review Committee in the Faculty of Medicine, University of Gezira, Gezira, Sudan. The study was carried out following the relevant ethical guidelines and regulations. All the respondents were clearly informed about the study purposes and were then asked for their consent to participate in the study by filling the questionnaire in the online form.

### Statistical analysis

We presented data in the form of mean and standard deviation (SD) for continuous variables, and frequencies and proportions for categorical variables. Data were categorized according to the level of knowledge and enrolment of participants in secondary research. Test of significant difference – Chi-square and Fisher exact test for categorical variables, and independent T-test for numerical variables – were used to find if there were significant differences in variables between participants with good versus poor knowledge and those who enrolled or did not enrol in secondary research. Logistic regression was used to assess the factors associated with good knowledge of SR. Variables with a p-value <0.05 were entered in the multivariate logistic regression. R software version 4.0.2 and SPSS 29 were used to conduct the analysis.

## Results

### Participants characteristics

A total of 13,060 medical students from nine Arab countries participated in this study, with a response rate of 89.7%. The largest proportion of students were from Sudan (32.1%), followed by Egypt (16.9%), Saudi Arabia (16.5%), Libya (7.8%), Syria (6.9%), Jordan (6.5%), Algeria (5.6%), Palestine (4.5%), and Yemen (3.2%). The mean age of the participants was 21.9±2.3 years, and 58.9% of the respondents were female. Most students were single (92.8%), living in urban areas

(68.5%), and enrolled in public (governmental) universities (77%). Of the total students, 51% were in their pre-clinical (basic) years, while 49% were in the clinical years of study. Moreover, 53% of the students reported having relatives in the healthcare field, with 35.9% having first-degree relatives and 25.4% having second-degree relatives (Table 1).

Out of the total sample, 5,848 students (44.8%) participated in research-related activities. However, only 31% of the students had heard of secondary research. Among the types of secondary research recognized, SR (20.8%) and meta-analysis (18.8%) were the most commonly identified. Only 25% of students had received training in secondary research, with 16.1% receiving that training as part of their medical school curriculum. Additionally, 13.5% of the participants reported having access to subscription-based databases (Table 2).

**Table 1. Baseline characteristics and difference in the level of knowledge for systematic review and practice in undergraduate medical students.**

| Variables | Overall, N = 13,060[1] | Level of knowledge | | p-value[2] | Enrolled or practiced in secondary research | | p-value[2] |
|---|---|---|---|---|---|---|---|
| | | Good, N = 563[1] | Poor, N = 12,497[1] | | Yes, N = 1,567[1] | No, N = 11,493[1] | |
| **Age, years** | 21.9 ± 2.3 | 22.6 ± 2.4 | 21.9 ± 2.3 | **<0.001** | 22.3 ± 2.4 | 21.9 ± 2.3 | **<0.001** |
| **Gender** | | | | 0.038 | | | **<0.001** |
| Female | 7,694 (58.9%) | 308 (54.7%) | 7,386 (59.1%) | | 827 (52.8%) | 6,867 (59.7%) | |
| Male | 5,366 (41.1%) | 255 (45.3%) | 5,111 (40.9%) | | 740 (47.2%) | 4,626 (40.3%) | |
| **Residence** | | | | 0.2 | | | **<0.001** |
| Rural | 4,109 (31.5%) | 190 (33.7%) | 3,919 (31.4%) | | 588 (37.5%) | 3,521 (30.6%) | |
| Urban | 8,951 (68.5%) | 373 (66.3%) | 8,578 (68.6%) | | 979 (62.5%) | 7,972 (69.4%) | |
| **Marital status** | | | | **<0.001** | | | **<0.001** |
| Divorced | 93 (0.7%) | 10 (1.8%) | 83 (0.7%) | | 26 (1.7%) | 67 (0.6%) | |
| Married | 756 (5.8%) | 66 (11.7%) | 690 (5.5%) | | 109 (7.0%) | 647 (5.6%) | |
| Single | 12,125 (92.8%) | 466 (82.8%) | 11,659 (93.3%) | | 1,421 (90.7%) | 10,704 (93.1%) | |
| Widowed | 86 (0.7%) | 21 (3.7%) | 65 (0.5%) | | 11 (0.7%) | 75 (0.7%) | |
| **Type of your university** | | | | 0.037 | | | 0.001 |
| Private | 3,006 (23.0%) | 150 (26.6%) | 2,856 (22.9%) | | 411 (26.2%) | 2,595 (22.6%) | |
| Public (Governmental) | 10,054 (77.0%) | 413 (73.4%) | 9,641 (77.1%) | | 1,156 (73.8%) | 8,898 (77.4%) | |
| **Year of Study** | | | | **<0.001** | | | **<0.001** |
| First-year | 1,532 (11.7%) | 38 (6.7%) | 1,494 (12.0%) | | 135 (8.6%) | 1,397 (12.2%) | |
| Second-year | 2,412 (18.5%) | 70 (12.4%) | 2,342 (18.7%) | | 208 (13.3%) | 2,204 (19.2%) | |
| Third-year | 2,666 (20.4%) | 110 (19.5%) | 2,556 (20.5%) | | 289 (18.4%) | 2,377 (20.7%) | |
| Fourth-year | 2,451 (18.8%) | 107 (19.0%) | 2,344 (18.8%) | | 310 (19.8%) | 2,141 (18.6%) | |
| Fifth-year | 2,015 (15.4%) | 109 (19.4%) | 1,906 (15.3%) | | 299 (19.1%) | 1,716 (14.9%) | |
| Sixth-year | 1,130 (8.7%) | 74 (13.1%) | 1,056 (8.5%) | | 179 (11.4%) | 951 (8.3%) | |
| Internship | 854 (6.5%) | 55 (9.8%) | 799 (6.4%) | | 147 (9.4%) | 707 (6.2%) | |
| **Current level in medical faculty** | | | | **<0.001** | | | **<0.001** |
| Basic | 6,666 (51.0%) | 233 (41.4%) | 6,433 (51.5%) | | 633 (40.4%) | 6,033 (52.5%) | |
| Clinical | 6,394 (49.0%) | 330 (58.6%) | 6,064 (48.5%) | | 934 (59.6%) | 5,460 (47.5%) | |
| **Having relatives in the healthcare field** | 6,925 (53.0%) | 347 (61.6%) | 6,578 (52.6%) | **<0.001** | 998 (63.7%) | 5,927 (51.6%) | **<0.001** |
| **First-degree relative** | 4,688 (35.9%) | 257 (45.6%) | 4,431 (35.5%) | **<0.001** | 754 (48.1%) | 3,934 (34.2%) | **<0.001** |
| **Second-degree relative** | 3,322 (25.4%) | 171 (30.4%) | 3,151 (25.2%) | **0.006** | 403 (25.7%) | 2,919 (25.4%) | 0.8 |
| **Ever participated in a research-related activity (e.g., class, workshop, or project)** | 5,848 (44.8%) | 440 (78.2%) | 5,408 (43.3%) | **<0.001** | 1,145 (73.1%) | 4,703 (40.9%) | **<0.001** |

[1]n (%); Mean ± SD.

[2]Pearson's Chi-squared test; Two Sample t-test; Fisher's exact test.

**Table 2. Level of knowledge about systematic review in undergraduate medical students.**

| Variables | Overall, N = 13,060[1] | Level of knowledge | | p-value[2] |
|---|---|---|---|---|
| | | Good, N = 563[1] | Poor, N = 12,497[1] | |
| **Heard about secondary research** | 4,052 (31.0%) | 467 (82.9%) | 3,585 (28.7%) | **<0.001** |
| **The known types of secondary research** | | | | |
| Systematic review | 2,713 (20.8%) | 317 (56.3%) | 2,396 (19.2%) | **<0.001** |
| Systematic review and meta-analysis | 2,456 (18.8%) | 310 (55.1%) | 2,146 (17.2%) | **<0.001** |
| Network meta-analysis | 992 (7.6%) | 207 (36.8%) | 785 (6.3%) | **<0.001** |
| Scoping review | 561 (4.3%) | 143 (25.4%) | 418 (3.3%) | **<0.001** |
| Narrative review | 814 (6.2%) | 169 (30.0%) | 645 (5.2%) | **<0.001** |
| **Received a training on systematic reviews** | 3,275 (25.1%) | 425 (75.5%) | 2,850 (22.8%) | **<0.001** |
| **The systematic reviews training source** | | | | |
| Medical school (in the curriculum) | 2,105 (16.1%) | 262 (46.5%) | 1,843 (14.7%) | **<0.001** |
| Physical extracurricular training | 791 (6.1%) | 157 (27.9%) | 634 (5.1%) | **<0.001** |
| Online extracurricular training | 1,112 (8.5%) | 202 (35.9%) | 910 (7.3%) | **<0.001** |
| Peer teaching | 521 (4.0%) | 111 (19.7%) | 410 (3.3%) | **<0.001** |
| **Have access to subscription databases (Paid databases) used in systematic review** | 1,758 (13.5%) | 240 (42.6%) | 1,518 (12.1%) | **<0.001** |
| **Questions about knowledge (percentage of correct answers)** | | | | |
| **Systematic review and meta-analysis provide highest level of evidence in research** | 3,536 (27.1%) | 340 (60.4%) | 3,196 (25.6%) | **<0.001** |
| **Prospero is the registry for systematic reviews protocols** | 829 (6.3%) | 341 (60.6%) | 488 (3.9%) | **<0.001** |
| **PRISMA is the reporting guideline for systematic reviews** | 985 (7.5%) | 348 (61.8%) | 637 (5.1%) | **<0.001** |
| **The programs used for conducting meta-analysis** | | | | |
| RevMan is used | 1,695 (13.0%) | 475 (84.4%) | 1,220 (9.8%) | **<0.001** |
| R software is used | 1,725 (13.2%) | 414 (73.5%) | 1,311 (10.5%) | **<0.001** |
| Stata is used | 1,581 (12.1%) | 404 (71.8%) | 1,177 (9.4%) | **<0.001** |
| SPSS version 23 is not used | 807 (6.2%) | 146 (25.9%) | 661 (5.3%) | **<0.001** |
| **The quality assessment tools used in systematic reviews** | | | | |
| Newcastle-Ottawa Scale is used | 1,585 (12.1%) | 440 (78.2%) | 1,145 (9.2%) | **<0.001** |
| PRISMA is not used | 1,160 (8.9%) | 250 (44.4%) | 910 (7.3%) | **<0.001** |
| Cochrane risk of bias tool is used | 1,836 (14.1%) | 452 (80.3%) | 1,384 (11.1%) | **<0.001** |
| NIH tool is used | 1,272 (9.7%) | 356 (63.2%) | 916 (7.3%) | **<0.001** |
| Meta-analysis is not used | 818 (6.3%) | 191 (33.9%) | 627 (5.0%) | **<0.001** |
| Could identify the forest plot | 1,247 (9.5%) | 329 (58.4%) | 918 (7.3%) | **<0.001** |
| **The statistical tests used to assess for heterogeneity across the studies in meta-analysis** | | | | |
| Chi-squared test is used | 2,397 (18.4%) | 476 (84.5%) | 1,921 (15.4%) | **<0.001** |
| T-test is not used | 1,215 (9.3%) | 261 (46.4%) | 954 (7.6%) | **<0.001** |
| I-squared test is used | 1,598 (12.2%) | 415 (73.7%) | 1,183 (9.5%) | **<0.001** |
| Inverse variance is not used | 979 (7.5%) | 223 (39.6%) | 756 (6.0%) | **<0.001** |
| **Random-effect model is used in the presence of significant heterogeneity** | 2,107 (16.1%) | 341 (60.6%) | 1,766 (14.1%) | **<0.001** |
| **Network meta-analysis is used to compare multiple treatments using direct comparisons of interventions within randomized controlled trials and indirect comparisons across trials based on a common comparator** | 954 (7.3%) | 284 (50.4%) | 670 (5.4%) | **<0.001** |
| **The score of knowledge (Total = 19)** | 2.2 ± 3.2 | 11.5 ± 1.8 | 1.7 ± 2.5 | **<0.001** |

[1]n (%); Mean ± SD.

[2]Pearson's Chi-squared test; Two Sample t-test; Fisher's exact test.

## Knowledge of SR among the students

The mean SR knowledge score was 2.2±3.2 out of a possible score of 19, indicating a poor overall level of knowledge. Only 563 students (4.3%) demonstrated good knowledge, while the overwhelming majority (95.7%) had poor knowledge (Table 2).

## Association of sociodemographic characteristics with the students' level of knowledge

Several sociodemographic factors were significantly associated with better SR knowledge, including being married or widowed, attending a private university, being in clinical years of study, and prior participation in research-related activities (p < 0.001). Using bivariate logistic regression, factors such as age, gender, type of university, year of education, clinical level, having a healthcare-related family member, and participation in research activities were found to be significantly associated with higher SR knowledge scores. Multivariate logistic regression analysis revealed that age (OR = 1.111, 95% CI: 1.069–1.154, p < 0.001) and previous participation in research activities (OR = 4.501, 95% CI: 3.650–5.551, p < 0.001) were the strongest predictors of good SR knowledge (Table 3).

**Table 3. Logistic regression for the factors associated with good knowledge of systematic review.**

| Variables | Univariate Logistic regression | | | | Multivariate Logistic regression | | | |
|---|---|---|---|---|---|---|---|---|
| | OR | 95% CI | | p-value | OR | 95% CI | | p-value |
| | | Lower | Upper | | | Lower | Upper | |
| **Age, years** | 1.122 | 1.086 | 1.159 | **< 0.001** | 1.111 | 1.069 | 1.154 | **< 0.001** |
| **Gender** | | | | | | | | |
| Male (Reference) | – | – | – | – | – | – | – | – |
| Female | 0.836 | 0.705 | 0.990 | **0.038** | 0.863 | 0.726 | 1.026 | 0.094 |
| **Residence** | | | | | | | | |
| Rural (Reference) | – | – | – | – | – | – | – | – |
| Urban | 0.897 | 0.750 | 1.072 | 0.233 | – | – | – | – |
| **Type of your university** | | | | | | | | |
| Public (Reference) | – | – | – | – | – | – | – | – |
| Private | 1.226 | 1.012 | 1.485 | **0.037** | 1.162 | 0.955 | 1.413 | 0.133 |
| **Year of Study** | | | | | | | | |
| First-year (Reference) | – | – | – | – | – | – | – | – |
| Second-year | 1.175 | 0.788 | 1.753 | 0.429 | – | – | – | – |
| Third-year | 1.692 | 1.164 | 2.460 | **0.006** | – | – | – | – |
| Fourth-year | 1.795 | 1.233 | 2.613 | **0.002** | – | – | – | – |
| Fifth-year | 2.248 | 1.545 | 3.273 | **< 0.001** | – | – | – | – |
| Sixth-year | 2.755 | 1.849 | 4.106 | **< 0.001** | – | – | – | – |
| Internship | 2.706 | 1.774 | 4.128 | **< 0.001** | – | – | – | – |
| **Current level in medical faculty** | | | | | | | | |
| Basic (Reference) | – | – | – | – | – | – | – | – |
| Clinical | 1.502 | 1.266 | 1.783 | **< 0.001** | 0.894 | 0.734 | 1.088 | 0.263 |
| **Having a relative in the healthcare field** | | | | | | | | |
| No (Reference) | – | – | – | – | – | – | – | – |
| Yes | 1.446 | 1.215 | 1.719 | **< 0.001** | 1.160 | 0.971 | 1.386 | 0.102 |
| **Ever participated in a research-related activity (e.g., class, workshop, or project)** | | | | | | | | |
| No (Reference) | – | – | – | – | – | – | – | – |
| Yes | 4.689 | 3.828 | 5.745 | **< 0.001** | 4.501 | 3.650 | 5.551 | **< 0.001** |

CI: Confidence Interval, OR= Odd ratio.

Knowledge of SR also varied significantly by country (p < 0.001). Students from Saudi Arabia had the highest proportion of "good knowledge" scores (10.8%), followed by students from Egypt (5.4%) and Palestine (5.3%). In contrast, students from Libya (0.6%), Sudan (2.2%), and Yemen (2.4%) had the lowest levels of knowledge (S1 Table).

### Awareness of medical databases

Students were asked about their awareness of biomedical databases. PubMed was the most commonly identified database (42.9%), followed by Google Scholar (36.2%) and Web of Science (15.0%) (Table 4).

### Secondary research skills

Regarding secondary research skills, searching online databases (13.3%) and screening titles/abstracts (10%) were the most frequently reported above-average skills among students. However, the least developed skills included conducting statistical meta-analysis (5.9%) and protocol registration (5.7%) (Table 5).

### Barriers to conducting SR

The barriers that the students faced towards conducting SR were the lack of knowledge about SRs (47.0%), lack of research exposure and opportunities (28.8%), lack of supervision and guidance (28.5%), lack of time (27.1%), difficulty of finding person conducting statistical meta-analysis (23.8%), lack of fund (23.2%), limited access to subscriptions database (23.0%), difficulty of finding experts authors in the SR (22.1%), inability to find research question (idea) (20.4%), inability to find full texts for articles (20.3%) and lack of interest (20.1%) (Table 6).

### The practice of SR

Of the total 13,060 students, only 1,567 students (12%) have participated in secondary research. The types of research projects the students mainly enrolled in were: SRs (62.3%), followed by SRs and meta-analysis (43.3%), then network

**Table 4. Medical databases known by undergraduate medical students.**

| Variables | N | Overall, N = 13,060[1] | Level of knowledge | | p-value[2] |
|---|---|---|---|---|---|
| | | | Good, N = 563[1] | Poor, N = 12,497[1] | |
| Known databases | 13,060 | | | | |
| PubMed | | 5,601 (42.9%) | 408 (72.5%) | 5,193 (41.6%) | **<0.001** |
| Google Scholar | | 4,732 (36.2%) | 293 (52.0%) | 4,439 (35.5%) | **<0.001** |
| Web of Science | | 1,958 (15.0%) | 276 (49.0%) | 1,682 (13.5%) | **<0.001** |
| Scopus | | 1,733 (13.3%) | 256 (45.5%) | 1,477 (11.8%) | **<0.001** |
| Science direct. | | 1,706 (13.1%) | 218 (38.7%) | 1,488 (11.9%) | **<0.001** |
| Cochrane central for randomized control trials | | 1,374 (10.5%) | 218 (38.7%) | 1,156 (9.3%) | **<0.001** |
| Clinicaltrial.gov | | 1,012 (7.7%) | 179 (31.8%) | 833 (6.7%) | **<0.001** |
| Health & Medical Complete (ProQuest) | | 1,010 (7.7%) | 130 (23.1%) | 880 (7.0%) | **<0.001** |
| Ovid | | 995 (7.6%) | 215 (38.2%) | 780 (6.2%) | **<0.001** |
| PsycINFO | | 842 (6.4%) | 162 (28.8%) | 680 (5.4%) | **<0.001** |
| African Journals OnLine (AJOL) | | 787 (6.0%) | 91 (16.2%) | 696 (5.6%) | **<0.001** |
| Health Collection (informit) | | 759 (5.8%) | 128 (22.7%) | 631 (5.0%) | **<0.001** |
| CINAHL (Cumulative Index to Nursing and Allied Health Literature) | | 635 (4.9%) | 117 (20.8%) | 518 (4.1%) | **<0.001** |
| Embase | | 17 (0.1%) | 11 (2.0%) | 6 (0.0%) | **<0.001** |
| Clinical Key | | 2 (0.0%) | 0 (0.0%) | 2 (0.0%) | >0.9 |

[1] n (%).

[2] Pearson's Chi-squared test; Fisher's exact test.

**Table 5. Rating skills of systematic review in undergraduate medical students.**

| | Poor | Limited experience | Average | Above average | Advanced |
|---|---|---|---|---|---|
| **Self-rated skills of systematic review** | | | | | |
| Formulating a question and its validation | 6,851 (52.5%) | 2,630 (20.1%) | 2,462 (18.9%) | 745 (5.7%) | 372 (2.8%) |
| Developing and writing a protocol (eligibility criteria, keyword search strategy) | 7,019 (53.7%) | 2,998 (23.0%) | 2,140 (16.4%) | 631 (4.8%) | 272 (2.1%) |
| Protocol registration | 7,863 (60.2%) | 2,494 (19.1%) | 1,981 (15.2%) | 469 (3.6%) | 253 (1.9%) |
| Creating search string | 7,432 (56.9%) | 2,594 (19.9%) | 2,188 (16.8%) | 584 (4.5%) | 262 (2.0%) |
| Searching online databases | 6,063 (46.4%) | 2,549 (19.5%) | 2,712 (20.8%) | 1,195 (9.2%) | 541 (4.1%) |
| Title and abstract screening | 6,839 (52.4%) | 2,587 (19.8%) | 2,332 (17.9%) | 871 (6.7%) | 431 (3.3%) |
| Full-text screening | 7,053 (54.0%) | 2,560 (19.6%) | 2,309 (17.7%) | 757 (5.8%) | 381 (2.9%) |
| Quality assessment of included studies | 7,431 (56.9%) | 2,627 (20.1%) | 2,096 (16.0%) | 619 (4.7%) | 287 (2.2%) |
| Data extraction | 6,864 (52.6%) | 2,656 (20.3%) | 2,354 (18.0%) | 811 (6.2%) | 375 (2.9%) |
| Statistical meta-analysis | 8,042 (61.6%) | 2,498 (19.1%) | 1,754 (13.4%) | 524 (4.0%) | 242 (1.9%) |
| Manuscript writing | 7,659 (58.6%) | 2,390 (18.3%) | 2,091 (16.0%) | 593 (4.5%) | 327 (2.5%) |

**Table 6. Perceived Barriers towards conducting systematic review in undergraduate medical students.**

| Variables | Overall, N = 13,060[1] |
|---|---|
| **The perceived barrier towards conducting systematic reviews** | |
| The lack of knowledge about systematic review | 6,136 (47.0%) |
| The lack of research exposure and opportunities | 3,756 (28.8%) |
| The lack of supervision and guidance | 3,719 (28.5%) |
| The lack of time | 3,544 (27.1%) |
| The difficulty of finding a person to conduct a statistical meta-analysis | 3,111 (23.8%) |
| The lack of fund | 3,036 (23.2%) |
| The limited access to subscription databases | 3,003 (23.0%) |
| The difficulty of finding experts authors in the systematic review | 2,890 (22.1%) |
| The inability to find a research question (Idea) | 2,664 (20.4%) |
| The inability to find full text for articles | 2,656 (20.3%) |
| The lack of interest | 2,619 (20.1%) |

[1] n (%).

meta-analysis (33.4%), and lastly, scoping reviews (28.6%). Of those who have participated in secondary research projects, 471 have published their work. The most common type of publication was SR (26.4%), followed by SR and meta-analysis (20.1%), network meta-analysis (16.2%), rapid reviews (13.3%), and scoping reviews (14.2%) (Table 7).

Among the 934 students actively involved in SR projects, PubMed was the most frequently used database (54.9%), followed by Google Scholar (37.9%), Web of Science (28.0%), Scopus (23.3%), ScienceDirect (21.6%), and the Cochrane Central Library for Randomized Controlled Trials (20.7%). The most frequent steps of SR that students took part in were formulating and validating the research question (47.5%), followed by searching online databases (34.3%), data extraction (30.8%), developing and writing the protocol (32.9%), and title and abstract screening (29.9%). Statistical meta-analysis was conducted using software such as RevMan (26.6%), SPSS (27.6%), Open Meta-Analyst (21.1%), the R package (20.0%), Comprehensive Meta-Analysis (CMA) software (20.6%), and Stata (20.3%) (Table 7).

**Table 7. Practice of systematic review in undergraduate medical students.**

| Variables | N | Overall, N = 13,060[1] |
|---|---|---|
| **Ever enrolled or participated in secondary research** | 13,060 | 1,567 (12.0%) |
| **Type of secondary research worked on** | 1,567 | |
| Systematic review | | 977 (62.3%) |
| Systematic review and meta-analysis | | 679 (43.3%) |
| Network meta-analysis | | 524 (33.4%) |
| Scoping review | | 448 (28.6%) |
| Rapid review | | 1 (0.1%) |
| **Ever published secondary research** | 1,567 | 471 (30.1%) |
| **Type of secondary research published** | 1,567 | |
| Systematic review | | 414 (26.4%) |
| Systematic review meta-analysis | | 315 (20.1%) |
| Network meta-analysis | | 254 (16.2%) |
| Scoping review | | 223 (14.2%) |
| Rapid review | | 208 (13.3%) |
| Others | | 220 (14.0%) |
| **The used databases in conducting systematic review** | 1,567 | |
| PubMed | | 860 (54.9%) |
| Google Scholar | | 594 (37.9%) |
| Web of Science | | 439 (28.0%) |
| Scopus | | 365 (23.3%) |
| Science direct | | 338 (21.6%) |
| Cochrane central for randomized control trials | | 325 (20.7%) |
| Ovid | | 286 (18.3%) |
| Clinicaltrial.gov | | 284 (18.1%) |
| Health & Medical Complete (ProQuest) | | 267 (17.0%) |
| African Journals Online (AJOL) | | 258 (16.5%) |
| PsycINFO | | 256 (16.3%) |
| Health Collection (informit) | | 250 (16.0%) |
| CINAHL (Cumulative Index to Nursing and Allied Health Literature) | | 213 (13.6%) |
| **The steps of the systematic review you worked on them** | 1,567 | |
| Formulate a question and its validation | | 744 (47.5%) |
| Develop and write protocol (eligibility criteria, keyword search strategy) | | 515 (32.9%) |
| Protocol registration | | 324 (20.7%) |
| Creating search string | | 429 (27.4%) |
| Searching online databases | | 538 (34.3%) |
| Title and abstract screening | | 469 (29.9%) |
| Full-text screening | | 449 (28.7%) |
| Quality assessment of included studies | | 364 (23.2%) |
| Data extraction | | 482 (30.8%) |
| Statistical meta-analysis | | 343 (21.9%) |
| Manuscript writing | | 389 (24.8%) |
| **The programs used in conducting meta-analysis** | 1,567 | |
| SPSS version 28 | | 433 (27.6%) |
| RevMan | | 417 (26.6%) |
| Open meta-analyst | | 330 (21.1%) |
| Comprehensive Meta-Analysis Software (CMA) | | 323 (20.6%) |

*(Continued)*

**Table 7.** (Continued)

| Variables | N | Overall, N = 13,060[1] |
|---|---|---|
| Stata | | 318 (20.3%) |
| R packages in R software | | 314 (20.0%) |

[1]n (%).

## Discussion

This study examined the popularity of SR and meta-analysis among undergraduate medical students in several Arab countries. It identified the essential variables affecting the level of knowledge among participants and the perceived barriers to practicing SR.

With 13,060 participating medical students, the mean knowledge score was 2.2 (SD = 3.2), reflecting a high percentage of students with a poor overall level of knowledge about secondary research (95.7%). This can be attributed to the fact that less than one-third of the participants have heard of secondary research (31%), and less than one-fifth are familiar with SRs (20.8%). The study findings showed a poor level of knowledge, which was found compatible with a study conducted among medical students of six Arab countries, which reported that out of 2,989 participants, 91.6% showed poor knowledge towards research as a whole [10]. Also, multivariate logistic regression revealed that age and participation in a research-related activity were significantly associated with good knowledge of secondary research.

This study identified a poor level of knowledge about SR, aligning with a previous study that reported a significant gap in research in Arab countries, identifying it as of poor quality [10]. One potential solution is involving medical students in research activities early in their education, such as through research assistantships, to foster evidence-based scientific thinking. Long-term approaches could focus on promoting physician-scientists to meet growing healthcare demands [10].

Among the participants, senior students demonstrated a better knowledge of SR than junior students, which is justified by their more extensive exposure to various aspects of the medical field. Also, being a married or widowed student and being a male, studying in a private university, in the fifth, sixth, or internship year, and at the clinical level, and participating in a research-related activity were factors related to good levels of knowledge about SR, which comes in agreement with a study that reported being a student in a top 1000 universities, having a father or a mother with a college degree and being in the clinical years were significantly associated with higher levels of knowledge towards research [10].

Participating in research was associated with a good level of knowledge about research in a previous study, assessed research analysis among medical residents in Saudi Arabia [11], and knowledge about research among undergraduate medical and dental students [12]. This confirms that medical students' involvement in research would enhance their understanding of research and SR. However, across Arab countries, Saudi Arabian students are leading the way with good levels of knowledge of SR, in contrast to Libyan students, who have the least level of knowledge. This finding differed from another study that assessed research knowledge in six Arab countries, which showed that Jordanian students had a high knowledge score about research, while Sudanese students had the lowest score [10]. These findings confirm that knowledge about research and SR differed across countries.

A previous study reported that SR and meta-analysis were the second most commonly conducted research in Egypt and the third in Jordan and Syria [10]. This could be explained by comparing awareness and attitudes towards medical research and the research fields in general, which provides insight into the educational system adopted among undergraduate medical students.

PubMed is widely used and gives free access to the MEDLINE (Medical Literature Analysis and Retrieval System Online, or MEDLARS Online) database [13]. Among the medical students participating in a study on EBM in Sudan, students reported that Google was the most commonly used search engine. In contrast, students enrolled in EBM training

had a higher level of using PubMed as a search engine [14]. Also, in Hungary, students of medicine and health sciences commonly used Google, followed by Wikipedia, and PubMed, and many students used PubMed [15]. The participants across countries in this study identified PubMed as the most reputable database, and this explains that familiarity with other databases and the variable types of SR is linked with the available opportunities to participate in SR projects as well as a stable fund resource, accessibility to databases, and availability of senior guidance, which is considered a problematic issue in the medical education field in the middle-east.

Regarding skills of SR, the most advanced skills reported in this study were searching for online databases (4.1%), followed by the title and abstract screening (3.3%), and the least below-average skill was in statistical meta-analysis (80.7%), followed by Protocol registration. Another previous Sudanese study about EBM reported limited skills and experience among medical students in locating professional literature, identifying relevant clinical questions, critical appraisal of scientific papers, and the ability to search online databases [14]. Generally, this explains that SR skills varied across participants from all countries.

In this study, the lack of knowledge was the most significant barrier facing medical students trying to adopt SR in practice, followed by the usual barriers such as a lack of time and funds, which are affected by the country's financial status and the institution. Additionally, a significant barrier was the lack of supervision and guidance, as well as the difficulty in finding authors with experience in SR, which raised concerns about a potentially vicious cycle of lack of expertise in SR and meta-analysis. These findings are supported by a study conducted in Saudi Arabia among senior medical students, who reported that a lack of professional supervision, inadequate training courses, and insufficient time and funding were the main obstacles to conducting research [16]. Another study among medical science students in Iran identified a lack of funding, inadequate guidance, time constraints, and a lack of knowledge and research skills as the most significant barriers to research [17]. This study, along with previous studies, has identified similar barriers that prevent participants from engaging in research and SR, which require a prompt solution to enhance and improve the knowledge and practice of SR.

Regarding the implementation of SR in practice, our results showed that only 12% of the participating medical students took on a role in such a project, with SR being more popular than meta-analysis and scope review. This low level of practice is consistent with the low level of knowledge about SR. A previous study in Saudi Arabia reported that the most popular research projects were case reports, retrospective, and prospective clinical studies, and cross-sectional and review articles [16]. Another study reported the most common type of studies to be cross-sectional, followed by case reports, and then SRs and meta-analysis [10]. These findings from previous studies confirm the low practice of SR in this study.

Due to the critical role that SRs play in evidence-based medicine, integrating SRs into medical school curricula is essential. Such training can equip medical students with skills in literature searching, data collection, and analysis, while also fostering critical research knowledge, scientific literacy, and improving their educational and career goals, which align with the goals of the World Federation of Medical Education [18].

This is the first cross-sectional study to tackle an issue concerning SR among medical students. The study is generalizable as it covered several Arab countries and included a large number of participants. The findings of this study will significantly aid in addressing the issue of knowledge and practice of SR among undergraduate medical students. However, this study didn't assess the quality of learning secondary research, which may affect the percentage of knowledge and Practice in each country. Additionally, non-probability sampling compromises the randomization of the sample and introduces a risk of selection bias.

## Conclusion

Knowledge and practice of SR processes among medical students were found to be low. Several barriers hindered undergraduate medical students from conducting SR. The findings of this study highlight the necessity to adopt SR and secondary research materials in curricula, as SR is becoming a postgraduate prerequisite worldwide. The findings of this study highlight the need to integrate SR and secondary research materials into medical school curricula and provide training

and capacity-building to enhance the medical workforce's ability to practice in an evidence-based manner. Peer-teaching projects offer valuable opportunities for practical learning. Faculty support is vital for students' research efforts, whether from formal duties or personal research involvement, as it profoundly impacts student engagement. We need to examine if students who published research did so independently or in collaboration with faculty to better understand the barriers and facilitators to conducting systematic reviews in different educational environments. Additionally, access to resources is a significant concern for undergraduate students. Academic institutions must fund library subscriptions to ensure students have unlimited access.

## Supporting information

**S1 Table. Includes the level of knowledge and practice of systematic review according to countries.**
(DOCX)

**S2 File. Included the data of the study.**
(XLSX)

## Acknowledgments

The authors would like to thank the research agency: "Sudan Analytics for Research and Statistics" (Links: https://www.facebook.com/Sudan.Analytics/; https://www.linkedin.com/company/76092484/; https://twitter.com/Sudananalytics; https://telegram.me/Sudananalytics) for their help in our research. Also, we would like to thank undergraduate medical students for participating in this study.

**Sudan Analytics Research Group team of collaborators:** Abdelrahim A Ahmed, Abdelrhman mohamed abdelrady, Abdelrhman Muwafaq Janem, Abdelrhman Waleed Elshenawy, Abdulghani Arja, Abdullah alshogaa, Abdullah Nasser Ali Hamilah, Abdurahim Mustafa Alaswad, Abeer S. El-Shawady, Abubakr Muhammad, Afaf Ahmad, Afnan Mousa Alhawsawi, Ahmad Ali Essa, Ahmed Khalid Yousif Mustafa, Al Zahraa Ali, Alaa Amr, Albushra A. Adam, Ali Alkhamis, Amna Altayeb Alhaj Ahmed, Aml Mostafa Farid, Amr Mahmoud Abdelraoof, Asma Ali alarabi, Asmaa H Alzabadiah, Awab Osman Ahmed Osman, Aya Ahmed Mohamed, Aya.M.abbas, Bashar Bazkke, DarElslam ELawad Osman, Deena faraj abdulali altarjami, Douaa Effat Belal, Ebtihal Mohammed Albasheir, Elsayed Hammad, Enaam alabed ahmed adam, Entisar Abdalla Zin Elabdein Ahmed, Esra Magzoub Ibrahem Ahmed, Esraa Salaheldin Ahmed Alfadul, Essraa ali alhudhairy, Faris Hamed, Ghadeer Ahmed Alhawsawi, Hagar K Elgazar, Haifa saif aldeen Alsaid humid, Haneen A. Banihani, Hassan Elamin Hassan Elamin, Hawa Essa Mohammed, Haya Deeb, Heba Alwaa, Hosam Khalouf, Hussein Mohammed Hussein Ghalib, ihcene Telaidji, Intisar Bubaker Owhida, Ismail Adam, Khadija Tariq Habib, Khaled Albakri, Khaled Tawfeeq, Khalid Elsir Idris, Khawla Alshahrani, Khetam Mohamed Saleh Adah, Khulod Ali Bokir, Kinda ahmed hardan, Lubna Hossam Aloufi, Lulyah almallah, Maazouz Bensalem, Maisa Nazzal Ayman Nazzal, Majid H Ali, Makhlouf Sofia, Manar Kahoul, Mawahib Elsheikh, Mawda Jumma, Mazin Mohammed Osman, Moath Almekhlafi, Moath Rushdi Atiani, Mobtahel Modather mohammed omer, Moftah Imhamed Moftah Himeda, Mohamed Abdelmonem, Mohamed Baklola, Mohamed Baraka, Mohamed Faroug Ali Yassin, Mohamed Gamal Eldow Abbas, Mohamed Ibrahem Baklola, Mohammad Abdulrahman Aljundi, Mohammad Saleh Alwajih, Mohammed Amir Rais, Mohmmad awd alsyed ahmad abbass, Monzer Elgamal, Malaz Omer Altegani Mohammed, Muhammad Sayed Muhammad Masoud, Mutwakel Ali Mohammed, Nagwa yassin Mohammad Taha, Nisreen Abufaris, Noon Isam Eldin Hasssan Hagmusa, Noura Ghazi Abu Lehia, Nourhan Elsamahy, Omar Abdulraheem Baqais, Omar Riffi, Omar S. Aldhafeeri, Omer A. Idrees, Qosay Sami Jabr, Raghad Nabeel Sadi Saleh, Rahaf mutlaq alotaibi, Rasha Nasser Hasnah, Rawan AbdAlwahd Mohammed Osman, Reema Saleh Zayed Alamri, Riham Mohammed Osman Mohammed, Roaa B. Albashir, Roua Arian, Ruwayda Abdulmutalib mawloud, Selema Nabeel Belgasem, Shahd T. Idais, Somayya Tabsho, Thekra Fawaz Albeshri, Turfa Moudarres, Waad Abu Elbeada, Wafaa Eshag

Moheyeldeen, Yousef Abdullah Aldreweesh, Ziad Alahmad, Rawa Naimat, Majedah A. Alakayleh, Shahed N. Abughanam, Mallak I. Alja'fari, Razan M. Altarawaneh, Ihab A. Alsalamin, Duoaa Albelal, Ahmad Aissa.

Affiliation: Sudan Analytics Research Group, Khartoum, Sudan.

The leading author is: Elfatih A. Hasabo (elfatih.ahmed.hasabo@gmail.com).

## Author contributions

**Conceptualization:** Elfatih A. Hasabo, Walaa Elnaiem, Alaa S. Ahmed, Azza E. A. Abdalla, Mohamed Sati Shampool Abdelgader, Khabab Abbasher Hussien Mohamed Ahmed, Ghassan Elfatih Mustafa Ahmed, Mohammed Mahmmoud Fadelallah Eljack.

**Data curation:** Elfatih A. Hasabo, Walaa Elnaiem, Alaa S. Ahmed, Azza E. A. Abdalla, Mohamed Sati Shampool Abdelgader, Khabab Abbasher Hussien Mohamed Ahmed, Ghassan Elfatih Mustafa Ahmed, Huda A Sherif, Omar Al Komi, Hajar Alkokhiya Aldare, Amira Yasmine Benmelouka, Afnan W. M. Jobran, Tayba A. Mugibel, MOHAMAD IMAD AL-KASSIH, Muhamad Zakaria Brimo Alsaman, Ahmed Aljabali, Mohammed Mahmmoud Fadelallah Eljack.

**Formal analysis:** Elfatih A. Hasabo, Mohammed Mahmmoud Fadelallah Eljack.

**Investigation:** Elfatih A. Hasabo, Walaa Elnaiem, Alaa S. Ahmed, Azza E. A. Abdalla, Mohamed Sati Shampool Abdelgader, Alamin Alfatih, Mohammed Mahmmoud Fadelallah Eljack.

**Methodology:** Elfatih A. Hasabo, Walaa Elnaiem, Alaa S. Ahmed, Azza E. A. Abdalla, Ghassan Elfatih Mustafa Ahmed, Mohammed Mahmmoud Fadelallah Eljack.

**Project administration:** Elfatih A. Hasabo, Walaa Elnaiem, Alaa S. Ahmed, Alamin Alfatih, Mohammed Mahmmoud Fadelallah Eljack.

**Resources:** Elfatih A. Hasabo, Walaa Elnaiem, Alaa S. Ahmed, Mohammed Mahmmoud Fadelallah Eljack.

**Software:** Elfatih A. Hasabo, Walaa Elnaiem, Mohammed Mahmmoud Fadelallah Eljack.

**Supervision:** Elfatih A. Hasabo, Mohammed Mahmmoud Fadelallah Eljack.

**Validation:** Elfatih A. Hasabo, Mohammed Mahmmoud Fadelallah Eljack.

**Visualization:** Elfatih A. Hasabo, Mohammed Mahmmoud Fadelallah Eljack.

**Writing – original draft:** Elfatih A. Hasabo, Walaa Elnaiem, Alaa S. Ahmed, Azza E. A. Abdalla, Khabab Abbasher Hussien Mohamed Ahmed, Ghassan Elfatih Mustafa Ahmed, Mohammed Mahmmoud Fadelallah Eljack.

**Writing – review & editing:** Elfatih A. Hasabo, Walaa Elnaiem, Alaa S. Ahmed, Azza E. A. Abdalla, Mohamed Sati Shampool Abdelgader, Khabab Abbasher Hussien Mohamed Ahmed, Ghassan Elfatih Mustafa Ahmed, Alamin Alfatih, Mohammed Mahmmoud Fadelallah Eljack.

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
