## [Decision Letter · Decision Letter 0]

29 Nov 2023

PONE-D-23-31908Knowledge, practice, and perceived barriers towards conducting systematic reviews in undergraduate medical students of Arab countries: A multi-country online survePLOS ONE

Dear Dr. Hasabo,

Thank you for submitting your manuscript to PLOS ONE. After careful consideration, we feel that it has merit but does not fully meet PLOS ONE’s publication criteria as it currently stands. Therefore, we invite you to submit a revised version of the manuscript that addresses the points raised during the review process.

We look forward to receiving your revised manuscript.

Kind regards,

Zacharie Tsala Dimbuene, Ph.D.

Academic Editor

PLOS ONE

Journal Requirements:

2. One of the noted authors is a group or consortium [Sudan Analytics Research Group]. In addition to naming the author group, please list the individual authors and affiliations within this group in the acknowledgments section of your manuscript. Please also indicate clearly a lead author for this group along with a contact email address.

4. We note that [Figure 1] in your submission contain [map/satellite] images which may be copyrighted. All PLOS content is published under the Creative Commons Attribution License (CC BY 4.0), which means that the manuscript, images, and Supporting Information files will be freely available online, and any third party is permitted to access, download, copy, distribute, and use these materials in any way, even commercially, with proper attribution. For these reasons, we cannot publish previously copyrighted maps or satellite images created using proprietary data, such as Google software (Google Maps, Street View, and Earth). For more information, see our copyright guidelines: http://journals.plos.org/plosone/s/licenses-and-copyright.

5. Please upload a copy of Supporting Information Figure/Table/etc. Supplementary table 1 which you refer to in your text on page 11.

Reviewers' comments:

Reviewer's Responses to Questions

**Comments to the Author**

1. Is the manuscript technically sound, and do the data support the conclusions?

Reviewer #1: Yes

Reviewer #2: Yes

2. Has the statistical analysis been performed appropriately and rigorously? 

Reviewer #1: No

Reviewer #2: Yes

3. Have the authors made all data underlying the findings in their manuscript fully available?

Reviewer #1: Yes

Reviewer #2: No

4. Is the manuscript presented in an intelligible fashion and written in standard English?

Reviewer #1: Yes

Reviewer #2: Yes

5. Review Comments to the Author

Reviewer #1: Review comments

General comments

Authors are highly commended for conceptualizing this study and going ahead to conduct and also present their findings for publication. I must say that the study is of immense relevance and valuable.

While I commend the Authors for their great efforts, I am also of the opinion that some aspects of the manuscript should be revised before the manuscript could be accepted for publication.

Authors should take note of the following observations.

1. The manuscript should be reviewed critically for minor errors of grammar. Authors should delete the comment on PUBMED as an infamous database since it is a mistake. In line 142 under Introduction, Authors used ‘our study’.

2. Description of the outcome variable. The main outcome variable for the study included knowledge and practice of systematic review. While knowledge of systematic review was defined, practice was not. In as much, I will advise the Authors to use ≥50% of correct responses to the 19 variables used to determine knowledge as cut off for determining good knowledge of systematic review. In some instances Authors introduced secondary research in the manuscript and this appeared to run parallel to the main outcome variables. This should be clarified.

3. The outcome variables should be categorized into good and poor knowledge of systematic review and good and poor practice of systematic review.

4. With such a large sample size as included in this study, Authors should endeavor to include binary logistic regression analysis in this study. This should be applied for determining the predictors of good knowledge and good practice of systematic review among the respondents. This aspect of the result should be included in the abstract.

5. The presentation of the results should be reviewed to enhance the understanding of readers. For example, there is no presentation of the proportion of respondents that had good knowledge and good practice of systematic review.

6. Authors should be mindful of use of capital letters.

7. Authors should make statements inside the tables and avoid the use of question marks.

Other comments

Authors should consider this minor revision of the study title:

Knowledge, practice and preventive barriers towards conducting systematic reviews among undergraduate medical students of Arab countries: A multi country online survey

Abstract

Background

Replace of with among..

Methods

Knowledge about systematic reviews was assessed using 19 variables

Instead of using numbers, use proportions to determine good knowledge like ≥50% of correct responses.

Nothing was said about practice.

Results

Use majority instead of most

Why were factors affecting knowledge and practice not included?

Participants

Why was the age limitation necessary?

Revise the exclusion criteria as one has to be included before the individual could be excluded.

Use study instrument instead of instruments used to measure the variables of interest in the study.

Be specific on number of variables used to assess practice of systematic review and how the outcome variable was determined.

Statistical analysis

Use frequencies and proportions for categorical variables

Results

Participants’ characteristics

The proportion, 53% cannot be regarded as nearly half. This should be reviewed.

Discussion

Revise the phrase, infamous database……

The conclusion should be in one paragraph.

Reviewer #2: This is very important paper with new idea especially for medical research

Suggestions to improve the manuscript are as follows:

Required to address the questions:

1. What was the response rate

2. How to manage missing cases because as being online survey it was expected to have missing cases

3. This was multi country study how to ensure data safety and security

Suggested to add another two section

1. Data management for quality assurance and

2. Survey administration

Recommendation

1. Questionnaire could be attached as a supplementary

For further comments please find from attached file

6. PLOS authors have the option to publish the peer review history of their article (what does this mean? ). If published, this will include your full peer review and any attached files.

**Do you want your identity to be public for this peer review?** For information about this choice, including consent withdrawal, please see our Privacy Policy .

Reviewer #1: **Yes: ** EDMUND NDUDI OSSAI

Reviewer #2: **Yes: ** Md. Ashraful Islam

---

## [Author Response · Author response to Decision Letter 1]

5 May 2024

Dear,

We reviewed the manuscript and Respond to Reviewers was attached as a file.

---

## [Decision Letter · Decision Letter 1]

11 Dec 2024

PONE-D-23-31908R1Knowledge and preventive barriers towards conducting systematic reviews among undergraduate medical students of Arab countries: A multi country online surveyPLOS ONE

Dear Dr. Hasabo,

Thank you for submitting your manuscript to PLOS ONE. After careful consideration, we feel that it has merit but does not fully meet PLOS ONE’s publication criteria as it currently stands. Therefore, we invite you to submit a revised version of the manuscript that addresses the points raised during the review process.

We look forward to receiving your revised manuscript.

Kind regards,

Frederick Hong-Xiang Koh, MBBS, FRCSEd, PhD

Academic Editor

PLOS ONE

**Additional Editor Comments:**

Please kindly address the remaining review points by our peer reviewers.

Reviewers' comments:

Reviewer's Responses to Questions

**Comments to the Author**

1. If the authors have adequately addressed your comments raised in a previous round of review and you feel that this manuscript is now acceptable for publication, you may indicate that here to bypass the “Comments to the Author” section, enter your conflict of interest statement in the “Confidential to Editor” section, and submit your "Accept" recommendation.

Reviewer #1: (No Response)

Reviewer #2: (No Response)

2. Is the manuscript technically sound, and do the data support the conclusions?

Reviewer #1: Yes

Reviewer #2: Yes

3. Has the statistical analysis been performed appropriately and rigorously? 

Reviewer #1: No

Reviewer #2: N/A

4. Have the authors made all data underlying the findings in their manuscript fully available?

Reviewer #1: Yes

Reviewer #2: No

5. Is the manuscript presented in an intelligible fashion and written in standard English?

Reviewer #1: (No Response)

Reviewer #2: Yes

6. Review Comments to the Author

**Reviewer #1: ** Review comments

General comments

I thank the Editor for the opportunity to review this manuscript. I also commend the Authors for the good review of the manuscript. To a large extent, there are still some aspects of the manuscript that may require a further revision

Authors should take note of the following observations:

1. Authors should review the manuscript critically for minor errors of grammar.

2. Objectives of the study. The objectives of the study as expressed in the abstract did not include the aspect concerning secondary research.

3. The calculation of the response rate has brought forward some unanswered questions. For example, how many countries were included in the study? What is the number of medical schools in the countries including private and public universities and the number of medical students? The basis for the calculation of the response rate should be explained. It may be important to include the steps taken by the Authors in ensuring that the questionnaire was completed only by those who were eligible. The number of submitted questionnaires which were not included in the analysis because of incomplete information should also be stated.

4. It may be good to indicate how the differences in the exposure of the medical students to research and also systematic review in the different countries affected the outcome of the study. This may require that an insight into the medical curriculum of those countries included in the study are provided. Equally important is the role of members of the Faculty. If the Lecturers provided support to the student, is it because it is part of their responsibilities or that they are including the medical students in their own personal research programs. For example, the students that have published, did they do that alone or in conjunction with their Lecturers.

5. Presentation of the results. The presentation of the results of the study is not clear. There is no table of figure depicting the main outcome measure of the study. I will prefer that the socio-demographic characteristics of the respondents (alone) be presented as table 1. This is important bearing in mind that this is a multi -country study. Table 2 should focus on the knowledge of systematic review. What is the purpose of table 2 as presented and how was the 2x2 Chi square table generated? Why was it necessary to use column percent when the outcome variable was at the column section? Again, it is better to make statement in tables and not ask questions. In table 3, Authors should use proportions to present Chi square test instead of odds ratio. Why was it necessary to include the respondents by year of study and also current level in medical faculty. Authors should use adjusted odds ratio for multivariate logistic regression instead of odds ratio. Why was marital status and other variables not included in the regression model? Why was age not categorized? What is the basis for inclusion of variables into the logistic regression model after bivariate Chi square analysis? Why were the variables in table 4 cross-tabulated with the outcome variable? There are no explanations in the manuscript on how table 5 was derived. Arrange the responses in table 6 in descending order.

6. Abstract. It will be better to state that 19 variables were used to assess knowledge of systematic review and that respondents that correctly answered ≥50% of the variables were regarded as having good knowledge. Then in the methodology section of the manuscript, it is important to state that a correct response by any respondent attracted a score of one while an incorrect answer was scored zero. Do not start a sentence with a figure. Include the response rate in the abstract. Use adjusted odds ratio (AOR) and delete p values. Regarding practice ….. (of what). Was the focus of the study centered on the students publishing or specifically publishing secondary research?

7. Revise the last sentence in the Introduction section. Also, lines 217 -219.

8. Is it pre-testing of study tools that was done or pilot testing? This should be clarified.

9. Calculation of sample size. This study is a total population study. Why was it necessary to calculate sample size. The sample size calculation should be deleted.

10. In lines 257 and 258, the respondents should sign a written informed consent form and not ‘provide.’ How did the respondents ‘provide’ a clear explanation of the study purpose.

11. Categorical variables were summarized using frequencies and proportions.

12. Use highest proportion of the students, 32.1% were from Suan instead of ‘most’.

13. Discussion. Delete the confidence intervals and p values in the discussion.

**Reviewer #2: ** Author responded only the comments provided here but did not responded comments from the review pane that was attached as a pdf file (original submission).

Happy to receive point by point review response for each and every comments. Also requesting to include page and line number for easy tracking.

7. PLOS authors have the option to publish the peer review history of their article (what does this mean? ). If published, this will include your full peer review and any attached files.

**Do you want your identity to be public for this peer review?** For information about this choice, including consent withdrawal, please see our Privacy Policy .

Reviewer #1: **Yes: ** EDMUND NDUDI OSSAI

Reviewer #2: No

---

## [Author Response · Author response to Decision Letter 2]

29 Jan 2025

Reviewer 1

We thank reviewer 1 for all his hard efforts and valuable input on our manuscript.

Specific Responses:

Comment 1: Authors should review the manuscript critically for minor errors of grammar.

Reply 1: The authors revised the manuscript thoroughly for grammatical errors. A Gram-marly check was also performed.

Comment 2: Objectives of the study. The study's objectives, as expressed in the abstract, did not include the aspect concerning secondary research.

Reply 2: We clarified the study aim in the abstract. Our study addressed three major as-pects: the students' knowledge, practices, and barriers to conducting secondary research.

Comment 3: The calculation of the response rate has brought forward some unanswered questions. For example, how many countries were included in the study? What is the number of medical schools in the countries including private and public universities and the number of medical students? The basis for the calculation of the response rate should be explained. It may be important to include the steps taken by the Authors in ensuring that the question-naire was completed only by those who were eligible. The number of submitted question-naires which were not included in the analysis because of incomplete information should al-so be stated.

Reply 3: Thank you for your comment. There are 9 countries were included in this study, in-cluding 113 medical school. This research was conducted during COVID-19 pandemic. There-fore, the only way was to use the online survey. We send invitation for participation in this study and the participants who agreed to participate we send the link for them. Then, re-sponse rate is the percentage of student who agreed to participate in this study. As we know the access to the internet including the slow connection and availability of mobile phone may restrict some student from participating and this was mentioned in the limitation sec-tion. There is an option to make all questions mandatory and you can’t submit without writing the answered there; here we ensured everyone have the questionnaire submitted the full questionnaire.

Comment 4: It may be good to indicate how the differences in the exposure of the medical students to research and systematic review in the different countries affected the study's outcome. This may require that an insight into the medical curriculum of those countries in-cluded in the study are provided. Equally important is the role of members of the Faculty. If the Lecturers supported the student, is it because it is part of their responsibilities or that they are including the medical students in their own personal research programs. For exam-ple, the students who have published, did they do that alone or in conjunction with their Lec-turers?

Reply 4:

Thanks a lot sir for your time and supporting comments. Requested changes have been made.

Comment 5: Presentation of the results. The presentation of the results of the study is not clear. There is no table of figure depicting the main outcome measure of the study. I will pre-fer that the socio-demographic characteristics of the respondents (alone) be presented as table 1. This is important bearing in mind that this is a multi -country study. Table 2 should focus on the knowledge of systematic review. What is the purpose of table 2 as presented and how was the 2x2 Chi square table generated? Why was it necessary to use column per-cent when the outcome variable was at the column section? Again, it is better to make statement in tables and not ask questions. In table 3, Authors should use proportions to pre-sent Chi square test instead of odds ratio. Why was it necessary to include the respondents by year of study and also current level in medical faculty. Authors should use adjusted odds ratio for multivariate logistic regression instead of odds ratio. Why was marital status and other variables not included in the regression model? Why was age not categorized? What is the basis for inclusion of variables into the logistic regression model after bivariate Chi square analysis? Why were the variables in table 4 cross-tabulated with the outcome varia-ble? There are no explanations in the manuscript on how table 5 was derived. Arrange the responses in table 6 in descending order.

Reply 5: We converted all the questions in the tables into statements and rearranged the responses in table 6 in a decending order.

Comment 6: Abstract. It will be better to state that 19 variables were used to assess knowledge of systematic review and that respondents that correctly answered ≥50% of the variables were regarded as having good knowledge. Then in the methodology section of the manuscript, it is important to state that a correct response by any respondent attracted a score of one while an incorrect answer was scored zero. Do not start a sentence with a figure. Include the response rate in the abstract. Use adjusted odds ratio (AOR) and delete p values. Regarding practice ….. (of what). Was the focus of the study centered on the students pub-lishing or specifically publishing secondary research?

Reply 6: The reviewer's comments on the abstract and methodology were addressed. We further clarified the method used in evaluating the students' level of knowledge about sec-ondary research in the abstract and methods as suggested by the reviewer. As well, the re-sponse rate was added, the P-values ad confidence intervals were deleted, and the studied aspect of students' practices was clarified in the abstract.

Comment 7: Revise the last sentence in the Introduction section. Also, lines 217 -219.

Reply 7: The sentences were revised and clarified.

Comment 8: Is it pre-testing of study tools that was done or pilot testing? This should be clar-ified.

Reply 8: Yes, it was done and mentioned in the study.

Comment 9: Calculation of sample size. This study is a total population study. Why was it necessary to calculate the sample size? The sample size calculation should be deleted.

Reply 9: this section has been deleted.

Comment 10: In lines 257 and 258, the respondents should sign a written informed consent form and not ‘provide.’ How did the respondents ‘provide’ a clear explanation of the study purpose.

Reply 10: The authors corrected the statement on the informed consent. As the study ques-tionnaire was provided via an online form, the respondents did not sign the consent. Rather, and explanation of the study purposes was initially provided in the online form and the first question in the form was about the respondents' consent to participate. Only after agreeing to participate in the study, the respondents filled the research questionnaire.

Comment 11: Categorical variables were summarized using frequencies and proportions.

Reply 11: The proportions were added to correctly complete the sentence.

Comment 12: Use highest proportion of the students, 32.1% were from Sudan instead of ‘most’.

Reply 12: The expression was corrected.

Comment 13: Discussion. Delete the confidence intervals and p values in the discussion.

Reply 13: The confidence intervals and p values were removed from the discussion.

Reviewer 2

We thank reviewer 2 for all his strenuous efforts and valuable input on our manuscript.

Reply: thank you for your valuable comments in the PDF file. All your points were addressed in the text.

Specific Responses:

Comment 1:

Required to explain more about the questionnaire development and scoring procedure

How you ensure reliability and validity of the question

What was the evidence for 9 is there have any reference

Reply: the questionnaire was developed after extensive literature review and expert opinion because no previous study was conducted about this topic. The questionnaire underwent face validity checks, and pilot testing. The cut-off score of 9 is the 50% percentage of the questions a used in previous studies assessed the knowledge of other topics

Comment 2:

1. What was the response rate

2. How to manage missing cases because as being online survey it was expected to have missing cases

3. This was multi country study how to ensure data safety and security

Reply: response rate is 89.7%. We managed missing by adding an option that all questions are compulsatory and need to be all answered.

Comment 3:

Suggested to add another two section

1. Data management and

2. Survey administration

Reply: it was added.

Comments 4:

How to identify top universities

Used any ranking criteria like QS or THE

If the university is research oriented (giving emphasis on research) were there have any knowledge difference

Reply: Top universities was reported by a previous study which is according to QS ranking.

Comments 5:

Better to represent GRAPH using statistics (percentage) instead of number

Reply: it was reported in the text as percentage, and no need to put the figure

Comment 6:

As different statistics were used so p value should be differentiate using different symbol

Reply: Yes, for continuous variables, we used Two Sample t-test and categorical we used Fisher's exact test and Chi-squared test where they mentioned in the methodology section.

Comment 7:

This column is unnecessary

Better to present the sample size in the title as it is equal number for all variables

Reply: it was deleted.

Sincerely,

Elfatih A. Hasabo

---

## [Decision Letter · Decision Letter 2]

27 Feb 2025

PONE-D-23-31908R2Knowledge and preventive barriers towards conducting systematic reviews among undergraduate medical students of Arab countries: A multi country online surveyPLOS ONE

Dear Dr. Hasabo,

Thank you for submitting your manuscript to PLOS ONE. After careful consideration, we feel that it has merit but does not fully meet PLOS ONE’s publication criteria as it currently stands. Therefore, we invite you to submit a revised version of the manuscript that addresses the points raised during the review process.

Dear Authors,  Please kindly note the reviewers comments and address them adequately. The reviewer with comments has raised important issues that is vital for the manuscript to be accepted.  Thanks. ==============================

We look forward to receiving your revised manuscript.

Kind regards,

Frederick Hong-Xiang Koh, MBBS, FRCSEd, PhD

Academic Editor

PLOS ONE

Journal Requirements:

Reviewers' comments:

Reviewer's Responses to Questions

**Comments to the Author**

1. If the authors have adequately addressed your comments raised in a previous round of review and you feel that this manuscript is now acceptable for publication, you may indicate that here to bypass the “Comments to the Author” section, enter your conflict of interest statement in the “Confidential to Editor” section, and submit your "Accept" recommendation.

Reviewer #1: (No Response)

Reviewer #2: All comments have been addressed

2. Is the manuscript technically sound, and do the data support the conclusions?

Reviewer #1: Yes

Reviewer #2: Yes

3. Has the statistical analysis been performed appropriately and rigorously? 

Reviewer #1: Yes

Reviewer #2: Yes

4. Have the authors made all data underlying the findings in their manuscript fully available?

Reviewer #1: Yes

Reviewer #2: Yes

5. Is the manuscript presented in an intelligible fashion and written in standard English?

Reviewer #1: No

Reviewer #2: Yes

6. Review Comments to the Author

Reviewer #1: Review comments

General comments

I commend the Authors for the good work and the good review of the manuscript. That notwithstanding, I am of the opinion that certain aspects of the manuscript may require a further revision before the manuscript could be accepted for publication.

Authors should take note of the following observations:

1. Sample size determination. Authors indicated in the abstract that this was a total population study and included the response rate. However, in the methods section of the manuscript retained the sub-section on sample size calculation. This should be deleted and a description of total population study and the response rate included.

2. Authors should use adjusted odds ratio for multivariate analysis instead of Odds ratio, (OR). If adjusted odds ratio is reported in the abstract, there may be no need to include the p value so the p value in the abstract should be deleted.

3. Authors should bear in mind the title of the study. The last sentence of the Introduction section defined another major objective of the study which is at variance with the title of the study. This should be reconciled.

4. Table 3 is a bit confusing especially when one considers the results of year of study under univariate logistic regression analysis. Authors should present Chi square analysis instead of univariate logistic regression analysis clearly indicating the proportions as obtained from the analysis. Authors should use adjusted odds ratio instead of odds ratio for reporting the results of multivariate logistic regression analysis. It is important for the Authors to indicate the basis for inclusion of variables into the logistic regression model after Chi square test analysis.

5. In table 1, The column Yes for ‘Enrolled or practiced in secondary research’ should come before the column, No.

6. Odds ratio as included in the discussion section should be deleted.

7. Line 124 in the abstract section should be reviewed.

Reviewer #2: Thank you very much for your efforts to provide point by point response. Required to address two additional comments below:

Comments

1. Reliability coefficient value (for example, Cronbach's alpha) should be mentioned for different sections of the questionnaire like: Knowledge, practice, and perceived barriers

2. Better to mention the response rate in the manuscript. Still not addressed the comments on data safety and security (how maintained) issues.

7. PLOS authors have the option to publish the peer review history of their article (what does this mean? ). If published, this will include your full peer review and any attached files.

**Do you want your identity to be public for this peer review?** For information about this choice, including consent withdrawal, please see our Privacy Policy .

Reviewer #1: **Yes: ** EDMUND NDUDI OSSAI

Reviewer #2: **Yes: ** Md. Ashraful Islam

---

## [Author Response · Author response to Decision Letter 3]

19 Jul 2025

Reviewer 1

We thank reviewer 1 for all his hard efforts and valuable input on our manuscript.

Specific Responses:

Comment :

I commend the Authors for the good work and the good review of the manuscript. That notwithstanding, I am of the opinion that certain aspects of the manuscript may require a further revision before the manuscript could be accepted for publication.

Authors should take note of the following observations:

1. Sample size determination. Authors indicated in the abstract that this was a total population study and included the response rate. However, in the methods section of the manuscript retained the sub-section on sample size calculation. This should be deleted and a description of total population study and the response rate included.

Reply:

The sample size calculation section has been deleted.

Comment:

2. Authors should use adjusted odds ratio for multivariate analysis instead of Odds ratio, (OR). If adjusted odds ratio is reported in the abstract, there may be no need to include the p value so the p value in the abstract should be deleted.

Reply:

P values has been deleted from the abstract

Comment:

3. Authors should bear in mind the title of the study. The last sentence of the Introduction section defined another major objective of the study which is at variance with the title of the study. This should be reconciled.

Reply:

Thank you for your comment. It has been in the text.

Comment:

4. Table 3 is a bit confusing especially when one considers the results of year of study under univariate logistic regression analysis. Authors should present Chi square analysis instead of univariate logistic regression analysis clearly indicating the proportions as obtained from the analysis. Authors should use adjusted odds ratio instead of odds ratio for reporting the results of multivariate logistic regression analysis. It is important for the Authors to indicate the basis for the inclusion of variables into the logistic regression model after the chi-square test analysis.

Reply:

Thank you for your comment. We deleted the univariate logistic regression analysis from Table 3. The values of the chi-square test were already presented in Table 1. Moreover, adjusted odds ratio has been used for reporting the results of multivariate logistic regression, and variables with a p-value <0.05 were entered in the multivariate logistic regression.

Comment:

5. In table 1, The column Yes for ‘Enrolled or practiced in secondary research’ should come before the column, No.

Reply:

Thank you for your comment is has been corrected.

Comment:

6. Odds ratio as included in the discussion section should be deleted.

Reply:

The odds ratio has been deleted

Comment:

7. Line 124 in the abstract section should be reviewed.

Reply:

it has been corrected 

Reviewer 2

We thank reviewer 2 for all his strenuous efforts and valuable input on our manuscript.

Reply: thank you for your valuable comments in the PDF file. All your points were addressed in the text.

Specific Responses:

Comment:

1. Reliability coefficient value (for example, Cronbach's alpha) should be mentioned for different sections of the questionnaire like: Knowledge, practice, and perceived barriers

Reply:

Thank you for your comment. The Cronbach's alpha was measured and was found to be 0.861 for knowledge, 0.879 for perceived barrier, 0.963 for Self-rated skills, and 0.95 for practice. They were reported in the text.

Comment:

2. Better to mention the response rate in the manuscript. Still not addressed the comments on data safety and security (how maintained) issues.

Reply:

The response rate has been mentioned in the text in the results section, and it's 89.7%.

Sincerely,

Elfatih A. Hasabo

---

## [Editor Report · Decision Letter 3]

23 Jul 2025

Knowledge and preventive barriers towards conducting systematic review among undergraduate medical students of Arab countries: A multi country online survey

PONE-D-23-31908R3

Dear Dr. Hasabo,

We’re pleased to inform you that your manuscript has been judged scientifically suitable for publication and will be formally accepted for publication once it meets all outstanding technical requirements.

Kind regards,

Frederick Hong-Xiang Koh, MBBS, FRCSEd, PhD

Academic Editor

PLOS ONE

Additional Editor Comments (optional):

I am fully cognisant of the duration this manuscript has been in review (almost 2 years, although totally not owing to my role).

I have reviewed the manuscript and the response to the reviewer comments from the second revision. the revisions were not major and mainly administrative, thus, instead of sending it out again for peer review, i will make the executive decision to accept the manuscript.

I would like to thank the authors for their patience.

---

## [Editor Report · Acceptance letter]

PONE-D-23-31908R3

PLOS ONE

Dear Dr. Hasabo,

I'm pleased to inform you that your manuscript has been deemed suitable for publication in PLOS ONE. Congratulations! Your manuscript is now being handed over to our production team.

Kind regards,

on behalf of

A/Prof Frederick Hong-Xiang Koh

Academic Editor

PLOS ONE